# fNIRS-Based Differences in Cortical Activation during Tool Use, Pantomimed Actions, and Meaningless Actions between Children with and without Autism Spectrum Disorder (ASD)

**DOI:** 10.3390/brainsci13060876

**Published:** 2023-05-29

**Authors:** Wan-Chun Su, McKenzie Culotta, Jessica Mueller, Daisuke Tsuzuki, Anjana Bhat

**Affiliations:** 1Department of Physical Therapy, University of Delaware, Newark, DE 19713, USA; wcsu@udel.edu (W.-C.S.); mckenzielculotta@gmail.com (M.C.); 2Biomechanics & Movement Science Program, College of Health Sciences, University of Delaware, Newark, DE 19713, USA; 3Department of Behavioral Health, Swank Autism Center, A. I. du Pont Nemours Children’s Hospital, Wilmington, DE 19803, USA; jka.mueller@gmail.com; 4Department of Information Science, Faculty of Science and Technology, Kochi University, Kochi 780-8520, Japan; tsuzukid@is.kochi-u.ac.jp; 5Interdisciplinary Neuroscience Graduate (ING) Program, Department of Psychological & Brain Sciences, University of Delaware, Newark, DE 19716, USA

**Keywords:** autism spectrum disorder, functional near-infrared spectroscopy, meaningless gestures, pantomime, praxis, tool use

## Abstract

Children with autism spectrum disorder (ASD) have difficulties with tool use and pantomime actions. The current study utilized functional near-infrared spectroscopy (fNIRS) to examine the neural mechanisms underlying these gestural difficulties. Thirty-one children with and without ASD (age (mean ± SE) = 11.0 ± 0.6) completed a naturalistic peg-hammering task using an actual hammer (hammer condition), pantomiming hammering actions (pantomime condition), and performing meaningless actions with similar joint motions (meaningless condition). Children with ASD exhibited poor praxis performance (praxis error: TD = 17.9 ± 1.7; ASD = 27.0 ± 2.6, *p* < 0.01), which was significantly correlated with their cortical activation (R = 0.257 to 0.543). Both groups showed left-lateralized activation, but children with ASD demonstrated more bilateral activation during all gestural conditions. Compared to typically developing children, children with ASD showed hyperactivation of the inferior parietal lobe and hypoactivation of the middle/inferior frontal and middle/superior temporal regions. Our findings indicate intact technical reasoning (typical left-IPL activation) but atypical visuospatial and proprioceptive processing (hyperactivation of the right IPL) during tool use in children with ASD. These results have important implications for clinicians and researchers, who should focus on facilitating/reducing the burden of visuospatial and proprioceptive processing in children with ASD. Additionally, fNIRS-related biomarkers could be used for early identification through early object play/tool use and to examine neural effects following gesture-based interventions.

## 1. Introduction

Besides their core social communication challenges and repetitive behaviors [1], children with ASD present with motor comorbidities, including poor motor planning/praxis and fine motor coordination [2,3,4,5,6] as well as impaired performance in skilled gestures, or dyspraxia [4,7,8,9]. Children with ASD have difficulties performing a range of functional gestures, including communicative/intransitive gestures (e.g., waving bye) as well as object-related/transitive gestures such as hammering a nail [7,10,11,12,13]. Motor skills, such as functional tool use, are essential for daily living, and studies focusing on the neural mechanisms of tool use could improve our understanding of developmental dyspraxia in children with ASD. Hence, using a peg-hammering paradigm, the current study investigated differences in behavioral errors and cortical activation during actual tool use, pantomime actions using an imagined object, and meaningless actions with similar movement kinematics between children with and without ASD. The neurobiomarkers of tool use identified in this study could also provide objective neural measures for monitoring intervention effects following interventions addressing dyspraxia.

### 1.1. Behavioral Framework for Tool Use

Tool use is an important ability in the evolution of primates that distinguishes humans from other species [14]. Through the development of tools, humans have preserved civilizations by countering obstacles from the forces of nature or other species and by creating tools for survival (e.g., agricultural tools, weaponry, and recreation tools). For individuals with difficulties using tools, problems not only occur during actual tool use but also during pantomimed actions on imagined tools [15]. When using actual tools or pantomiming tool use, multiple sensorimotor and cognitive processes are involved [16]. First, one needs to perceive the characteristics and the opportunities provided by the tool, including the potential motor actions on the tool (e.g., affordance/grasp-ability), and the mechanical actions between the tool and the object (e.g., cut-ability, hammer-ability) [17,18]. According to the embodied manipulation-knowledge hypothesis, one must refer to the action semantic system that stores functional knowledge about the tool (i.e., what the tool is used for) as well as the action knowledge system that stores movement information required to use the tool (i.e., ways or actions for grasping the tool) [15]. According to this approach, difficulties in identifying tools and referring to functional knowledge result in “content errors” (errors in how a tool is used, e.g., banging a toothbrush as if it were a knife) while impaired action knowledge, motor control, and action errors result in “production errors” (atypical movement production, e.g., holding a hammer at the distal end, putting too much or too little force on a nail, or missing the nail, and so on) [19]. A more recent technical-reasoning hypothesis suggests that tool-use actions are based on mechanical knowledge and require a specific form of causal and analogical reasoning for the user to anticipate outcomes and apply reasoning to novel situations [18,20,21]. Lastly, during the execution of a tool-use action, one must be aware of movement errors using feedforward and feedback control processes before and during movement execution, respectively, to make the necessary plans/corrections [16,22,23]. Figure 1 shows the behavioral framework for tool use.

Pantomiming tool use is considered more cognitively demanding than actual tool use, as it additionally requires movement imagery and heavily relies on our internal models to predict sensorimotor consequences and specify movement commands to achieve a successful movement outcome [24]. One common error that apraxic adults make during pantomimed tool use is the body part-for-tool error, wherein they use a body part to represent the imagined tool (e.g., cutting motions using the index and middle fingers to represent the use of scissors) [25]. It is suggested that by using body parts to represent a tool, the cognitive demands of imagining the tool’s presence are reduced [26]. 

### 1.2. Neural Mechanisms Underlying Tool Use 

Lesion studies on stroke patients have offered insights into our understanding of the neural mechanisms underlying gestural performance. Compared to patients with right hemispheric strokes, individuals with left hemispheric strokes are more likely to suffer from apraxia, indicating a left-lateralized brain network for tool-related gestural performance [27]. The left-lateralized neural network has been further confirmed through functional magnetic resonance imaging (fMRI) studies showing greater left than right hemispheric activation during pantomime and tool use actions, regardless of the hand used [28,29]. A meta-analysis of 70 neuroimaging contrasts found a tool-use network related to tool recognition, naming, and action in the left inferior parietal lobule (IPL) and ventral premotor cortex, a tool-identification network in the bilateral occipito-temporal cortices, and a tool-manipulation (action-related) network in the left superior parietal lobule and dorsal premotor cortex [30]. Studies supporting the manipulation-knowledge hypothesis comprise a tool-use network based on the recognition of a tool (action semantic systems) and the planning/execution of tool-related movements (action knowledge system; Figure 1). For example, the left IPL region, including the supramarginal gyrus, is activated when recognizing/naming tools and when planning/executing tool-use actions [30]. An fMRI study found significantly greater activation of the left parietal lobe (including IPL, superior parietal lobe, and supramarginal gyrus) when planning and executing actual tool use compared to control manipulations, suggesting that the parietal lobe encodes tool-specific information [29]. Even during pantomiming without the presence of an actual tool, the left IPL regions are activated irrespective of the hand used for pantomime actions [28,31]. Besides the left parietal lobe, the frontal and temporal regions are also important for recognizing, planning, and executing skilled gestures [30,32]. The posterior temporal cortex, including the superior temporal sulcus (STS) and middle temporal gyrus (MTG), is important for identifying tools and determining their associated actions [32]. Using positron emission tomography (PET) scans, Kaltenbach et. al. (2003) found greater activation of the posterior MTG regions when healthy adults made judgments about the functions of manipulable objects compared to those of non-manipulable objects, suggesting its important role in processing the functional knowledge associated with tools [33]. On the other hand, the frontal lobes, including the middle and inferior frontal gyrus (MIFG) and premotor cortex, are said to encode the motor representations required for planning and executing tool-related manipulations, as they are more active when grasping and manipulating tools [30,34,35]. On the other hand, studies supporting the technical-reasoning hypothesis confirmed the network by distinguishing between visuospatial and technical reasoning, where technical reasoning requires causal reasoning in physical contexts and is unique in tool use (Figure 1) [20,21,36]. For example, using fMRI, Fischer et al. (2016) found greater activation of the left frontal and parietal regions in participants viewing physical vs. social interactions and making physical vs. non-physical judgments [36]. Moreover, Federico et al. (2022) found that the cortical thickness of the left but not right IPL (especially the anterior part of the supramarginal gyrus; PF region) correlated with technical-reasoning performance [37]. This suggests that the left PF is involved in the technical reasoning of tool use, while the right PF is more involved in visuospatial processing [37]. Taken together, the studies supporting both hypotheses suggest a left-lateralized frontal-parietal network that is important for performing actual and pantomimed tool-use actions. 

Differences in the neural mechanisms between actual and pantomimed tool use are not well understood [38]. However, most researchers agree that pantomiming tool-use actions is more cognitively demanding, as it requires imagining tool use and the movement consequences induced by tool-related actions [24]. Several fMRI studies compared neural activation between actual and pantomimed tool use and found inconsistent results. For example, Ohgami et al. (2004) found a greater activation of the right supramarginal gyrus when healthy adults pantomimed tool-use actions [31]. The right supramarginal gyrus is important for analyzing joint positions and processing proprioceptive inputs and, therefore, might be more activated during pantomimed tool use [39,40]. While Imazu et al. (2007) found greater left-IPL activation, Lausberg et al. (2015) found greater activation in the left, middle, and superior temporal gyrus (MSTG) during pantomimed tool use compared to actual tool use [41,42]. The current study will add to our understanding of the neural mechanisms underlying tool-related gestures, as we examine the differences in cortical activation and behavioral errors between naturalistic actual and pantomimed tool use in children with and without ASD.

### 1.3. Dyspraxia in Children with ASD

Dyspraxia has long been viewed as a common co-occurring difficulty in children with ASD, as they consistently show atypical intransitive and transitive gestures involving actual/pantomimed tool use [4,5,7,9,10,11,13]. Specifically, children with ASD make more spatial errors (i.e., atypical joint coordination and amplitude of gesture), body part-for-tool errors (i.e., using body parts to represent a tool), and content errors (i.e., using tools in ways that they are not supposed to be used) compared to their typically developing (TD) peers when performing tool-related gestures [7]. Basic motor control challenges were found to correlate with praxis performance in children with ASD, suggesting that their difficulties in performing skilled gestures might be due to basic motor difficulties [11]. More specifically, difficulties in visuomotor control may be contributing to the dyspraxia patterns seen in children with ASD [4,5,9,11,13]. Besides basic motor control, children with ASD showed atypical visuospatial-processing performance that might hinder technical reasoning, an important process in tool use [43,44]. Some have suggested that deficits in imitating gestures might be a core symptom of children with ASD and directly relate to their social communication difficulties [2,3,45,46]. However, others have reported generalized difficulties in gestural performance in children with ASD with intransitive gestures as well as gestures performed during tool use, imitation, and upon verbal commands [7,9,13]. In the current study, we examined cortical activation when performing tool-related gestures upon verbal command, i.e., engaging in tool use or pantomimed tool use, to rule out any effects of imitation on gestural performance. 

### 1.4. ASD-Related Atypical Cortical Activation during Tool Use and Pantomimed Tool Use 

Several fMRI and electroencephalogram (EEG) studies have suggested atypical neural activation patterns in children with ASD when engaging in pantomiming tool-related gestures [47,48,49]. A resting-state fMRI study in children with ASD found a reduced connectivity between the bilateral inferior parietal cortices and dorsal premotor cortices, and this atypical connectivity correlated with the children’s praxis performance [47]. When processing the meaning of observed gestures, an fMRI study found that children with ASD had reduced sensitivity to movement excursions (subtle or exaggerated movements) with reduced activation in the right posterior STS region [48]. During pantomimed tool use, an EEG study found that children with ASD showed reduced motor/premotor beta (18–22 Hz) event-related desynchronization (ERD) and reduced left parietal alpha ERD (7–13 Hz) compared to TD peers, suggesting atypical motor imagery [49]. Previous fMRI and EEG studies have suggested that atypical gesture processing and motor imagery might underlie ASD-related difficulties in tool-related gestural performance. However, even though fMRI and EEG are considered gold-standard neuroimaging tools, they have their own limitations. For instance, fMRI requires participants to lie still in the scanner, which may not be suitable for children with ASD, while EEG has poor spatial resolution and cannot detect the spatial origin of functional activation [50]. Because of these limitations, no study has investigated ASD-related differences in cortical activation during actual tool-use actions in an upright position. It is questionable whether previous fNIRS and EEG findings could be generalized to naturalistic, everyday gesture tasks. Moreover, the previous studies interpreted their results based on grounded, embodied hypotheses (e.g., the manipulation-knowledge hypothesis) without considering the reasoning process during tool use, which will be considered in this study.

### 1.5. Aims and Hypotheses of This Study

Functional near-infrared spectroscopy (fNIRS) is a non-invasive neuroimaging technique with a greater temporal resolution than fMRI and a greater spatial resolution than EEG [50,51]. More importantly, its post-processing steps accommodate movement artifacts and the system only constrains the subject by a cap—unlike fMRI, which requires the participants to lay still in a scanner bore—making it an ideal neuroimaging tool for children with ASD [50,51]. Using fNIRS, our own research group reported consistent patterns of cortical activation in typical adults and school-age children with ASD that differ from children with ASD when perceiving, performing, and imitating reaching for objects, whole-body sway, and cooperative joint actions, which we plan to use as intervention response measures following embodied social interventions [52,53,54,55,56,57,58]. Given the advantage of fNIRS, the current study expanded the gesture paradigm to investigate neural activity during actual tool use. Therefore, the objective of the current study was to investigate cortical activation in children with and without ASD during actual tool use, pantomimes of tool-use actions, and when performing meaningless actions with similar movement kinematics. Moreover, we aimed to correlate tool-related cortical activation with standardized praxis measures and ASD severity in children with ASD. More importantly, we utilized both traditional, grounded manipulation-knowledge and recent technical-reasoning hypotheses to interpret our findings (Figure 1). We hypothesized that TD children would show left-lateralized cortical activation, whereas children with ASD would show a lack of left lateralization when engaging in tool-related gestures (actual or pantomime). In terms of group differences, children with ASD would show atypical activation patterns in the frontal, parietal, and temporal regions across all conditions (actual tool use, pantomimed tool use, and meaningless actions) compared to TD children.

## 2. Materials and Methods

### 2.1. Participants

Thirty-one children with and without ASD (mean age ± SE: ASD group: 11.14 ± 0.95, 9 males and 5 females; TD group: 10.82 ± 0.69, 11 males and 6 females, Table 1) participated in the study. There were no significant age or sex differences between groups (*p* > 0.05). We recruited the participating children by posting announcements with various ASD advocacy groups, local schools, and community centers as well as word of mouth. Screening interviews were scheduled to obtain demographic information (i.e., age, sex, ethnicity, and socioeconomic status using the Hollingshead Four-Factor Index of Socioeconomic Status [59]) and to confirm eligibility before a child participated in the study. The children with ASD were included if they held a professionally confirmed ASD diagnosis that was supported by school records, such as an individualized education plan for ASD-related services, medical/neuropsychological records from a psychiatrist or clinical psychologist based on the Autism Diagnostic Observation Schedule (ADOS), and/or clinical judgment [60]. We also screened for a social communication delay, or a score at or above 12 on the Social Communication Questionnaire (SCQ; Table 1) [61]. Children with ASD were excluded if they had any sensory and/or behavioral issues that prevented them from wearing the fNIRS cap and engaging in seated activities for about 30 minutes. TD children were age- and sex-matched to children with ASD and were excluded for any neurological or developmental disorders/delays or for having a family history of ASD. A clinical psychologist (i.e., the 3rd author) independently confirmed the diagnosis of ASD using the Autism Diagnostic Observation Schedule, 2nd edition (ADOS; average ADOS score ± SE = 18.17 ± 1.84) [60]. In addition, parents completed the Vineland Adaptive Behavioral Scales, 2nd edition (VABS) [62], as a measure of their child’s adaptive functioning. The Coren’s handedness survey provided a measure of the child’s handedness (Table 1) [63]. All study procedures were carried out in accordance with the study procedures approved by the University of Delaware Institutional Review Board (UD IRB, Study Approval number: 930721). All children gave their verbal assent, and their parents/legal guardians signed written consent forms before study participation. Written parental and experimenter permission/consent to use their pictures for this publication was received as well.

### 2.2. Study Procedures

We used the postural praxis (SIPT-PP) and bilateral motor-coordination subtests (SIPT-BMC) [64] of the Sensory Integration and Praxis Tests to study children’s praxis performance. During the SIPT-PP test, each child sat in front of and mirrored the actions of the adult tester (e.g., if the tester moved their right hand, then the child moved their left hand). Seventeen standardized upper-limb, body, and hand poses were shown by the tester, and the child was given the opportunity to demonstrate their response. During the SIPT-BMC test, the tester demonstrated a series of hand- and foot-tapping motions, and the child was asked to repeat the action sequences in the same rhythm and order. There were 10 hand-tapping and 4 foot-tapping movements in total. All behavioral assessments were videotaped for further behavioral coding.

Before the fNIRS data collection, the testing procedures were explained to the child, and the child was given the opportunity to practice the actual and pantomimed tool-related gestures. Each child remained seated at a desk for all three conditions: (a) During the hammer condition, a hammer and peg set (arranged in two rows, 4 pegs per row) was presented to the child. The child was asked to pick up the hammer and hit the pegs from left to right in the front and back rows (Figure 2a). (b) During the pantomime condition, the hammer and peg set were removed, and the child pretended to pick up a hammer and then hammer imaginary pegs in the same order as in the hammer condition (Figure 2b). (c) In the meaningless condition, the child was asked to tap the air 8 times (4 times in the front and 4 times in the back) to mimic the kinematics of the hammering motion (Figure 2c). The tester made sure that the child understood the instructions before starting the experiment. During data collection, an fNIRS cap embedded with two 3 × 3 probe sets was placed on the child’s head. The child was asked to complete a total of 18 trials (6 trials per condition that were randomized across the entire session; see the order in Figure 2d). Each trial included a 10second pre-stimulation, a 16-second stimulation period, and a 16-second post-stimulation period. The pre- and post-baseline periods were used to pick up signals related to baseline drifts in the fNIRS signal and to allow the hemodynamic response to return to baseline before the start of the next trial. During the baseline periods, each child was asked to remain still and observed a crosshair on the front wall.

### 2.3. fNIRS Data Collection

The Hitachi fNIRS system (ETG-4000, Hitachi Medical Systems, Inc; recording rate: 10 Hz) was used to record hemodynamic changes. The fNIRS cap was embedded with two 3 × 3 probe sets, covering the bilateral parietal, frontal, and temporal regions. To ensure consistency in cap placement, the middle column of each probe set was aligned with the tragus point of each ear and the lowest border was placed just above the ear (the T3 position of the international 10–20 system, Figure 3a,b) [65]. Each fNIRS probe set consisted of 5 infrared emitters and 4 receivers, separated at a distance of 3 cm, and arranged in an alternating fashion (Figure 3a,b). The emitter emitted two wavelengths of infrared light (695 and 830 nm) through the skull, creating a banana-shaped arc that reached the cortical region in approximately the middle of the two probes, creating a channel. The infrared light was absorbed by tissue and traveled through the banana-shaped arc before it was detected by the receiver. The attenuation of infrared light was used to calculate changes in the concentrations of oxygenated (HbO_2_) and deoxygenated hemoglobin (HHb) chromophores using the modified Beer–Lambert law [50]. When a brain region was more activated, an increase in HbO_2_ concentration and a decrease in HHb concentration compared to the baseline period was seen [50]. We used E-Prime presentation software (version 2.0) to trigger the Hitachi fNIRS system based on a randomized block design experiment. In addition, a camcorder synchronized with the fNIRS system was used to record videos of the children and the tester during the experiment.

### 2.4. Spatial Registration Approach

The spatial registration approach uses the 3D positions of standard cranial landmarks and probe locations to obtain the brain regions covered underneath each fNIRS channel. Children were asked to sit still while the procedure was conducted. The Polhemus motion tracking system was used to register cranial landmarks (nasion, inion, left and right tragus points of each ear, and the Cz point, based on the international 10–20 system) and the 3D locations of each probe with respect to a reference coordination system. Using the anchor-based spatial registration method developed by the 4th author, the cranial coordinates were used to calculate the Montreal Neurological Institute (MNI) coordinates of each channel [66]. The MNI location of each channel was referenced against structural information obtained from an anatomical database developed by Okamoto et al. (2004) [67], and brain regions underneath each channel were labeled using the LONI Probabilistic Brain Atlas (LPBA) to provide estimates of channel positions within a standardized, 3D brain atlas [68]. The channel position data from all children were averaged and then assigned to the three regions of interest (ROI). The three ROIs included (i) the middle and inferior frontal gyrus (MIFG; consists of channels 1, 3, 6, 8 on the left and channels 14,17,19, 22 on the right; Figure 3c,d), (ii) the middle and superior temporal gyrus (MSTG, consists of channels 10,11,12 on the left and channels 20, 23, 24 on the right; Figure 3c,d), and (iii) the inferior parietal lobe (IPL, consists of channels 2, 4, 5, 7 on the left and channels 13, 15, 16, 18 on the right; Figure 3c,d). Channels 9 and 21 have been excluded as a result of spatial uncertainty. Detailed information on the channel assignments is presented in Appendix A.

### 2.5. Video Coding for SIPT-PP and SIPT-BMC

The first author and a student researcher, blinded to each child’s diagnosis, coded the number of errors each child made during the SIPT-PP and SIPT-BMC tests. For the SIPT-PP subtest, an error was assigned to an action if the child used incorrect hand orientations, joints, or body parts or demonstrated an insufficient or exaggerated range of motion compared to the tester. For the SIPT-BMC subtest, an error was assigned to an action if the child failed to repeat the action in the same sequence, rhythm, or pace (faster/slower) as the tester. We established inter-rater (SIPT-PP: 98% and SIPT-BMC: 94%) and intra-rater (SIPT-PP: 99% and SIPT-BMC: 94%) reliability between the two coders using 20% of the dataset to obtain strong reliability before a blinded coder coded the remaining data.

### 2.6. Processing Cortical-Activation Data

We developed custom MATLAB (The Mathworks Inc., Natick, MA) codes to analyze the .csv output from ETG-4000 using functions from open-source software, such as Hitachi POTATo [69] and Homer-2 [70], to analyze the fNIRS data. As described in our previous papers [52,53,54], the following methods were used to process the fNIRS data: (i) the data were band-pass filtered (between 0.01 and 0.05 Hz) to remove higher and lower frequency noise, including respiration, heart rate, etc.; (ii) the wavelet method was used to remove motion artifacts [70,71]; (iii) the GLM method (including Gaussian base functions and a 3rd order polynomial drift regression) was used to estimate the hemodynamic response function [70]; (iv) linear trends between the pre-and post-stimulation baseline were subtracted from values in the stimulation period to account for baseline drifts [69]; (v) the HbO_2_ and HHb values per trial were averaged across all time frames in the stimulation period; (vi) the averaged HbO_2_ and HHb values for each channel belonging to the same ROI were further averaged across channels within each ROI to reduce the number of comparisons across conditions; and (vii) HbO_2_ signal data were reported in this study, as the signal-to-noise ratio is known to be higher in HbO_2_ signals than in HHb signals [71].

### 2.7. Data Exclusion for Cortical Activation Data

Based on video screening, cortical activation data were excluded if (a) the child did not follow instructions for the given condition, or (b) no cortical activation data were obtained (flat line) as a result of poor probe–scalp contact. A student researcher, blinded to the study hypotheses, screened through the videos to identify trials in which the child failed to follow instructions or engage in appropriate actions. In the TD group, 1.5% of the data (0% in the hammer, 1.8% in the pantomime, and 2.7% in the meaningless conditions) were excluded, while in the ASD group, 4.7% of the data (2.4% in the hammer, 7.1 in the pantomime, and 4.7 in the meaningless conditions) were excluded. Additionally, the first author conducted a visual screening of plotted fNIRS data and excluded trials with no fNIRS signal. In the TD group, 3.6% of the data (4.2% in the hammer, 3.5% in the pantomime, and 3.2 % in the meaningless conditions) were excluded, whereas in the ASD group, 9.9% of the data (10.6% in the hammer, 9.5 in the pantomime, and 9.7 in the meaningless conditions) were excluded because of signal quality.

### 2.8. Statistical Analyses

We used *t*-tests to compare differences in praxis errors (SIPT-PP and SIPT-BMC error scores) between groups. Cortical activation differences were compared using a repeated-measures ANOVA in IBM SPSS (SPSS Inc.) with the within-group factors of condition (hammer, pantomime, meaningless), hemisphere (left, right), and region of interest (ROI: MIFG, MSTG, IPL), a between-group factor of group (children with and without ASD), and the controlling factors of BOT-2 manual dexterity and IQ score. Greenhouse–Geisser corrections were applied when the cortical activation data violated the sphericity assumption based on the Mauchly’s test of sphericity. To account for multiple comparisons during post-hoc analyses, we used the false discovery rate (FDR) method to adjust the cutoff point of the *p*-threshold for significant differences [72]. Specifically, the unadjusted *p*-values were ranked from the smallest to the largest, and statistical significance was declared if the unadjusted *p*-value was less than the *p*-value threshold. The *p*-thresholds were determined by multiplying 0.05 by the ratio of the unadjusted *p*-value rank to the total number of comparisons (*p*-threshold for ith comparison = 0.05 × i/n, where n = total number of comparisons). The FDR method is often used in fNIRS studies to adjust for multiple comparisons [72].

## 3. Results

### 3.1. Praxis Performance

The children with ASD had more praxis errors in the postural praxis (SIPT-PP) and bimanual coordination subtests (BMC) of the Sensory Integration and Praxis Test compared to the children without ASD (SIPT-PP total error (mean ± standard error (SE)): TD = 17.9 ± 1.7; ASD = 27.0 ± 2.6; SIPT-BMC total error: TD = 8.4 ± 1.5; ASD = 28.1 ± 5.2; *ps* < 0.01).

### 3.2. Cortical Activation

The group–condition–hemisphere–region four-way repeated ANOVA revealed a significant main effect from hemisphere (F (1.0, 195.0) = 4.1, *p* < 0.05), two-way interactions between group and hemisphere (F (1.0, 195.0) = 19.7, *p* < 0.001) and between condition and region (F (3.5, 680.6) = 4.1, *p* < 0.01), a three-way interaction among condition, hemisphere, and region (F (3.6, 3.8) = 2.8, *p* < 0.05), and a four-way interaction among group, condition, hemisphere, and region (F (3.6, 708.0) = 3.24, *p* < 0.05). Note that the main and interaction effects reported above did not covary with children’s intelligence quotient (IQ) or Bruininks–Oseretsky Test of Motor Proficiency (BOT-2) manual dexterity scores. Figure 4a,b shows the color-coded mean oxygenated hemoglobin (HbO_2_) concentration during hammer, pantomime, and meaningless conditions in children with and without ASD. The means and SEs of HbO_2_ concentrations are presented in Appendix A, while the results of the post-hoc analyses are presented in Appendix A.

#### 3.2.1. Hemispheric Differences

The TD children without ASD showed consistent left-lateralized cortical activation (left > right) across all regions of interest (ROIs) during all conditions (nine out of nine ROI comparisons, *ps* < 0.001, Figure 5a), while children with ASD did not show left-lateralized activation for five out of the nine ROIs (*ps* > 0.05, Figure 5b). However, the ASD groups exhibited left-lateralized activation in the MSTG and IPL ROIs for the hammer condition (*ps* < 0.01) as well as in the MIFG ROI for the pantomime and meaningless conditions (*ps* < 0.05, Figure 5b).

#### 3.2.2. Condition-Related Differences

TD children showed the greatest cortical activation during the pantomime condition and the lowest activation during the hammer condition (i.e., the task with lowest need for imagery, Figure 6a). Specifically, the TD children showed greater activation of the right, middle, and inferior frontal gyri (MIFG), MSTG, and IPL regions during pantomime condition compared to the hammer condition (*ps* < 0.01, Figure 6a). In addition, they also showed greater right-MSTG activation during the meaningless condition compared to the hammer condition (*p* < 0.01, Figure 6a). Children with ASD had condition-related differences in cortical activation in the right MSTG region only, with greater right-MSTG activation found during the pantomime condition compared to during the hammer and meaningless conditions (*ps* < 0.01, Figure 6b).

#### 3.2.3. ASD-Related/Group Differences

ASD-related differences were similar across all three conditions, with the greatest differences found in the hammer condition (three group differences), followed by the meaningless condition (two group differences) and pantomime condition (one group difference). Specifically, during the hammer condition, the children with ASD showed a lower cortical activation of the left MSTG region (*p* < 0.001) and a greater activation of the right MIFG and IPL regions compared to the TD children without ASD (*ps* < 0.01; Figure 7a). During the pantomime condition, the children with ASD showed greater right-IPL activation than the TD children without ASD (*p* < 0.01, Figure 7b). Lastly, during the meaningless condition, the children with ASD showed reduced left-MSTG and greater right-IPL activation compared to the TD children (*ps* < 0.01; Figure 7c).

### 3.3. Correlations between Cortical Activation and Praxis Performance

Significant correlations were found between cortical activation and praxis performance in both groups. However, TD children showed more significant correlations between SIPT-PP/SIPT-BMC errors and cortical activation in the left than the right hemisphere (number of significant correlations: left hemisphere—10, right hemisphere—1), while the children with ASD showed a similar number of significant correlations in each hemisphere (number of significant correlations: left—5, right—7; Table 2). Specifically in the TD children, the SIPT-PP error was correlated with left-MIFG activation during the hammer and pantomime conditions, and with left-STG and right-IPL activation during the meaningless condition (R-values ranged from −0.283 to 0.311; *ps* < 0.001; Table 2). They also showed significant correlations between SIPT-BMC scores and left-MIFG and -IPL activation during hammer and meaningless conditions, as well as left-MIFG, -MSTG, and -IPL activation during the pantomime condition (R-values ranged from −0.296 to −0.426, *ps* < 0.001; Table 2). In children with ASD, the SIPT-PP errors were correlated to the activation of the bilateral MIFG and right IPL regions during the hammer and meaningless conditions, as well as the right MIFG region during the pantomime condition (R-values ranged from −0.257 to 0.434, *ps* < 0.001, Table 2). Moreover, children with ASD showed significant correlations between SIPT-BMC errors and cortical activation of the right IPL region during the hammer condition, the left MSTG region during the pantomime condition, and the left MSTG and bilateral IPL ROIs during the meaningless condition (R-values ranged from 0.317 to 0.543, *ps* < 0.001, Table 2).

### 3.4. Correlations between Cortical Activation and ADOS Scores in Children with ASD

Significant correlations were found between cortical activation and ADOS scores in children with ASD, especially in the RRB domain. The Autism Diagnostic Observation Schedule Social Affect (ADOS-SA) score significantly correlated with left-MSTG activation during the pantomime condition (R = 0.310, *p* < 0.001; Table 3).

The Autism Diagnostic Observation Schedule Repetitive Behaviors (ADOS-RRB) score significantly correlated with activation in the left IPL and right MIFG regions during the hammer condition, the left MSTG ROI during the pantomime condition, and the left MSTG and right IPL regions during the meaningless condition (R-values ranged from 0.311 to 0.360, *ps* < 0.001; Table 3). The ADOS total score correlated with left-IPL activation during the hammer condition and left-MSTG activation during the pantomime condition (R = 0.343 and 0.361 respectively, *ps* < 0.001; Table 3).

### 3.5. Correlations between Cortical Activation and VABS Scores

Cortical activation in the left MIFG and bilateral MSTG and IPL ROIs was associated with VABS scores in children with and without ASD. The associations differed between children with and without ASD, indicating differences in the neural networks used for tool-related gestures. VABS composite scores and socialization, communication, and daily living subdomain scores significantly correlated with right-MIFG activation (but not left-MIFG activation) in both groups (positive R for right-MIFG activation in the TD group for eight out of twelve comparisons = 0.24 to 0.49, *ps* < 0.05; Appendix A). However, weaker associations were noted between right-MIFG activation and VABS scores in the ASD group (R for right-MIFG activation in the ASD group = −0.25 to 0.26). Multiple of the VABS socialization, communication, and daily living scores significantly correlated with left- and right-MSTG activation for multiple gestural conditions in both groups (positive R for hammer = 0.23 to 0.34, *ps* < 0.05, and negative R for pantomime and meaningless = −0.19 to −0.26, *ps* < 0.05; Appendix A). Lastly, the VABS scores significantly correlated with left- and right-IPL activation for multiple gestural conditions in both groups (positive R for left-IPL activation in the TD group = 0.25 to 0.27, *ps* < 0.05, and negative R for left-IPL activation in the ASD group = −0.22 to −0.34, *ps* < 0.05; Appendix A).

## 4. Discussion

Children with ASD have impaired gestural performance or dyspraxia, as seen by greater errors in actual and pantomimed tool use [4,7,8,9,10]. In this study, we examined the neural mechanisms related to tool-use performance in children with ASD, given its value in learning and performing everyday skills. Previous fMRI and EEG studies have reported atypical activation/connectivity in the frontal, temporal, and parietal regions when children with ASD observed and pantomimed tool-use gestures [47,48,49]. However, no study has examined the ASD-related differences in functional activation during actual tool use and how activation differs between actual and pantomimed tool use. Using a peg hammering-paradigm, the present study compared functional activation between children with and without ASD during actual tool use, pantomimed tool use, and meaningless actions with similar kinematics. Consistent with the previous literature, children with ASD exhibited impaired praxis performance compared to children without ASD, as they made far more errors during the standardized praxis tests. In terms of cortical activation, children both with and without ASD exhibited left-lateralized functional activation across all three gestural conditions and a greater right-hemispheric activation during pantomimed rather than actual tool use. However, the aforementioned patterns differed between children with and without ASD, with the activated ROIs differing between groups. In terms of group differences in cortical activation, children with ASD exhibited reduced left-MSTG and greater right-MIFG and -IPL activation during actual tool use compared to children without ASD. Children with ASD also exhibited greater right-IPL activation during pantomimed tool use, and reduced MSTG activation and greater right-IPL activation during meaningless gestures compared to children without ASD. Functional activation patterns associated with gestural performance in children with and without ASD correlated with their standardized praxis performance. Additionally, functional activation patterns in children with ASD were also associated with their ASD severity. Lastly, functional activation patterns in both groups were associated with the children’s social communication and daily skills functioning; however, the relations differed between groups.

### 4.1. Impaired Praxis Performance in Children with ASD

Using the SIPT-PP and SIPT-BMC subtests, we found increased praxis errors in children with ASD compared to their TD peers. Our findings are consistent with past studies that have reported impaired praxis performance in children with ASD using various standardized tests [3,4,9,13,73]. The SIPT-PP subtest focused more on the spatial aspects of copying poses (i.e., an error was noted if the child copied the pose shown to them using the wrong joint/body part or with insufficient or exaggerated actions compared to the tester). During a hand gesture task, Salowitz et al. (2013) reported lower accuracy in hand shapes, orientation, and number of constituent limb movements in children with ASD compared to children without ASD [74]. Using a charades game, Fourie et al. (2020) found a decrease in the overall quality of gestural performance and atypical hand positions in children with ASD compared to those without ASD [48]. Children with ASD might have an atypical/rigid movement repertoire and poor motor coordination that could lead to difficulties in copying the spatial aspects of actions and postures [8,9,13,22]. In contrast, the SIPT-BMC subtest focused more on motor coordination during rhythmic actions. For example, an error was noted if the child failed to repeat the movements in the correct sequence, did not follow the correct movement rhythm, or moved faster/slower than the tester. Children with ASD are known to have difficulties with upper- and lower-limb coordination that are linked to their social communication and functional abilities [3,4,5,6]. Moreover, they show greater variability during synchronized finger tapping, drumming, marching, and clapping, indicating difficulties with movement timing [3,75,76]. The SIPT-PP and SIPT-BMC error scores were significantly correlated with cortical activation during all conditions in both children with and without ASD, suggesting that similar neural networks were being utilized for generalized praxis as well as gestural performance during the peg-hammering paradigm. Additionally, the activation findings in the present study could be generalized to a broader range of praxis tasks.

### 4.2. Left-Lateralized Activation in Children without ASD and More Bilateral Activation in Children with ASD during Gestural Performance

Past studies have indicated a left-lateralized cortical network that is important for tool-related actions [30,77]. In addition, in the present study, children were instructed to engage in gesture tasks using their right hand, contributing to greater left-hemispheric activation in the contralateral than the ipsilateral hemisphere (i.e., left-lateralized activation). Therefore, it is not surprising to see consistent left-lateralization in children without ASD in all ROIs and all conditions. In contrast, children with ASD showed greater bilateral cortical activation in multiple ROIs, with MIFG being bilaterally active during the hammer condition and MSTG and IPL being bilaterally active during pantomime and meaningless conditions. Moreover, the SIPT-PP and SIPT-BMC error scores mostly correlated with cortical activation in the left hemisphere in TD children, whereas in children with ASD, the error scores correlated to cortical activation in both hemispheres. These findings further suggest a less left-lateralized/more bilaterally activated neural network during tool-related actions in children with ASD. It has been postulated that during childhood, there is greater interhemispheric integration during the performance of tool-use gestures [47]. Hence, it is possible that the lack of left-hemispheric lateralization during tool use is indicative of a delayed trajectory in the development of tool-related neural networks in children with ASD. Alternatively, children with ASD are also known to have atypical hemispheric specialization, specifically, reduced left-hemispheric dominance (i.e., social communication, language, and motor-related symptoms) and relatively greater right-hemispheric activation, which may be compensatory in nature [78]. In a resting-state fMRI study, children with ASD had a weaker connectivity between the bilateral parietal lobes and premotor cortices compared to children without ASD, and the children’s atypical connectivity in these regions correlated with their praxis performance [47]. Our past fNIRS study also found a lack of left-lateralization, or greater bilateral activation, in children with ASD compared to those without ASD [54]. Overall, compared to TD children, children with ASD may be recruiting different neural circuits (greater bilateral, inferior parietal, and superior temporal activation) when performing pantomimed and meaningless gestures to account for the greater need for action imagery/planning.

### 4.3. Greater Right-Hemispheric Activation during Pantomimed Actions in Children with and without ASD

In terms of condition-related differences in gestural performance, we found greater right-hemispheric activation during the pantomime condition than during the hammer condition in children with and without ASD. Previous fMRI studies comparing neural activity between pantomime and actual tool use have reported inconsistent findings. For example, Imazu et al. (2007) found greater left-IPL activation during pantomime compared to the actual use of chopsticks [41], while Lausberg et al. (2015) found greater activation of the left, middle, and superior temporal gyrus when pantomiming multiple tool-use actions compared to actual tool use [42]. These inconsistencies might arise from the specificity or complexity of the tool used (e.g., greater fine motor control in using chopsticks compared to peg hammering) and/or the type of instruction (e.g., verbal or visual cues, etc.). Generally, the majority of findings from fMRI studies support the hypothesis that greater cognitive demands are required during pantomimed tool use compared to actual tool use. Compared to actual tool use, pantomimed tool-use actions require the imagination of a tool, planning of joint positions, and proprioception of movement consequences induced by the tool-related actions [24]. Several regions of the right hemisphere, including the right STS and supramarginal gyrus, are sensitive to the visual and proprioceptive feedback of movements and, therefore, might show differentiated activation during pantomime compared to actual tool use [39,40,79]. More studies involving actual tool use are needed to understand the similarities and differences between the neural mechanisms underlying pantomime and actual tool use.

### 4.4. Atypical Activation of the Parietal, Frontal, and Temporal Networks during Actual and Pantomimed Tool Use in Children with ASD

Consistent with our hypothesis, children with ASD were found to exhibit atypical cortical activation in the frontal, temporal and parietal regions when performing tool-related actions. Specifically, they exhibited greater right-IPL activation for all three types of gestures, greater right-MIFG activation during actual tool use, and reduced MSTG activation during actual and pantomimed tool use. Additionally, children with ASD had different neural networks activated during the actual and pantomimed tool use compared to those without ASD, which were associated with their social communication and daily living skill performance as well as their ASD severity.

#### 4.4.1. Intact Left IPL but Hyperactivation of Right IPL in Children with ASD

We found that children with ASD showed a typical activation of the left IPL but hyperactivation of the right IPL region during all three types of gestures. The left IPL is critical for tool-related actions, including recognizing/naming tools, planning/executing tool-use actions, and pantomiming tool-related actions [28,29,30]. In addition, the left IPL, particularly the anterior portion of the supramarginal gyrus, is said to be important for technical reasoning [37]. While we did not observe group differences in left-IPL activation, we found greater activation in the right IPL in children with ASD compared to those without ASD. The right supramarginal gyrus (part of the IPL region) is known to play an important role in visuospatial processing [37]. It is also activated when healthy adults use their body parts to represent tools [31] or when they perform meaningless gestures, indicating its importance in analyzing limb positions and processing proprioceptive information [39,40]. Greater right-IPL activation in children with ASD might reflect their atypical performance in processing visuospatial and proprioceptive information during tool use [39,40,43,44]. Children with ASD might have intact technical reasoning (indicated by the typical left-IPL activation) but atypical visuospatial and proprioceptive processing (indicated by the hyperactivation of the right IPL), leading to insufficient tool-use performance.

#### 4.4.2. ASD-Related Hyperactivation of Right MIFG Region during Actual Tool Use

Besides the hyperactivation of the right IPL region, ASD-related hyperactivation of the right MIFG was also seen during the hammer condition. The IFG and premotor cortices are activated when grasping and manipulating tools and have been suggested to encode the motor representations required for planning and executing tool-related actions [30,34,35]. The hammer condition was the only condition during which a tool was presented. The hyperactivation of the right MIFG region during the hammer condition might indicate the recruitment of different neural circuits (greater bilateral, middle, and inferior frontal gyri) when performing actual tool use to account for their difficulties with goal-directed action planning/praxis.

#### 4.4.3. ASD-Related Hypoactivation of Left MSTG When Performing Tool Use and Meaningless Gestures

Lastly, compared to their TD peers, children with ASD were found to exhibit reduced MSTG activation during actual tool use and when performing meaningless gestures. The MSTG region is important for identifying tools and determining associated purposeful actions [32]. A positron emission tomographic study found greater activation of the posterior MTG region when healthy adults made judgments about the functions of manipulable objects compared to judgements about non-manipulable objects, suggesting its important role in processing the functional knowledge associated with tools [33]. The reduced MSTG activation in children with ASD might reflect their difficulties in processing functional affordances and in using this information towards the planning of similar actions within the context of performing tool-related gestures.

### 4.5. Limitations and Future Directions

In the current fNIRS study, we were able to identify ASD-related neurobiomarkers related to actual and pantomimed tool use. However, this relatively small sample study has its own limitations. We were constrained by the lack of depth resolution of fNIRS and were unable to cover activity in the subcortical regions. Secondly, we used a 24-channel probe set to reduce the weight of the fNIRS probes placed on the children’s heads, foregoing whole brain coverage. Our peg-hammering paradigm was relatively structured in that each child was told to perform certain actions per condition following practice and did not engage in spontaneous gesture production during social interactions. To account for these limitations, our currently ongoing study uses the largest fNIRS probe set for maximum coverage as well as a naturalistic charades game to compare communicative and tool-related gestures in children with and without ASD.

## 5. Conclusions

Using standardized praxis tests, the current study reconfirmed the atypical patterns of skilled gestural performance in children with ASD. Moreover, children with ASD’s praxis performance was correlated to their cortical activation during actual and pantomimed tool use. The ASD-related neurobiomarkers we identified when participants performed tool-related gestures include a pattern of bilateral activation in multiple cortical regions (MSTG and IPL) related to tool use, hyperactivation of the right IPL and MIFG regions during actual and/or pantomimed tool use, and hypoactivation of the MSTG regions. More importantly, our results suggest intact technical reasoning (i.e., typical left-IPL activation) but atypical visuospatial and proprioceptive processing (i.e., hyperactivation of the right IPL) during tool use in children with ASD. Lastly, pantomimed tool use required greater cortical demands compared to actual tool use, as it evoked greater right-hemispheric activation in children with and without ASD. Clinically speaking, we should facilitate children with ASD in their visuospatial and proprioceptive processing performance by challenging their sensorimotor integration with, for example, visuomotor skills training [80]. During intervention, we should first provide a tool in hand when asking children with ASD to produce tool-related gestures and gradually increase the challenge by asking them to pantomime gestures (e.g., pretending to be a soldier waving a sword, etc.). There could be value in engaging in charade-like games to help them practice pantomimed gestures. On the other hand, since anxiety and other behavioral symptoms might affect children’s tool-use performance and training outcomes, techniques such as deep pressure may help to manage ASD-related symptoms [81]. We also recommend recording fNIRS-based cortical activation longitudinally during tool use in infancy to identify early neurobiomarkers of ASD and before and after gesture-based interventions to record neural changes. The neurobiomarkers identified in the current study could be used as objective measures of an intervention response [58].

## Figures and Tables

**Figure 1 brainsci-13-00876-f001:**
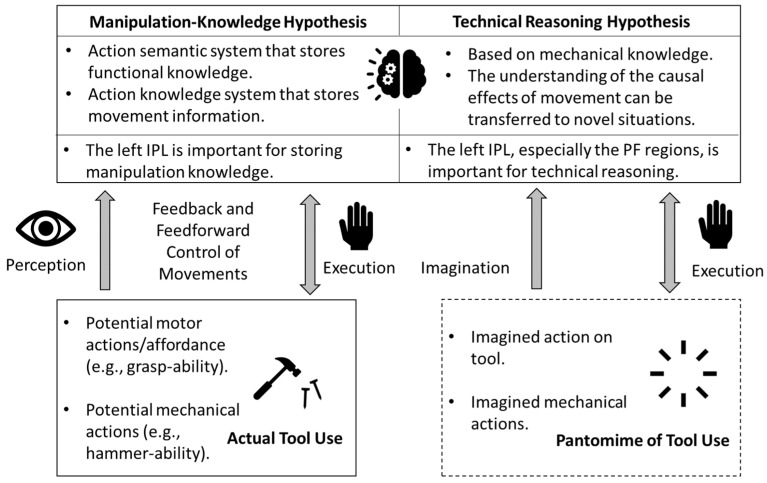
Behavior framework and potential neural mechanisms for tool use.

**Figure 2 brainsci-13-00876-f002:**
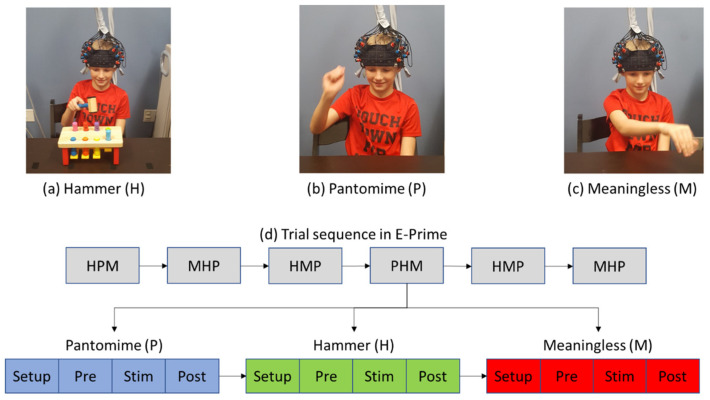
Experimental setup for the hammer (**a**), pantomime (**b**), and meaningless (**c**) conditions, as well as the task sequence (**d**). Written permission for publication of participant pictures has been received.

**Figure 3 brainsci-13-00876-f003:**
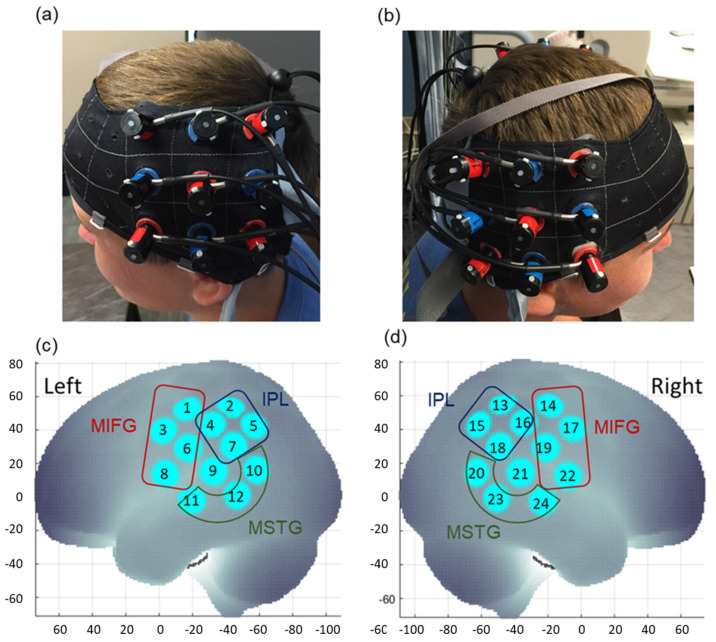
Probe placement (**a**,**b**) and spatial registration output (**c**,**d**). Written permission for publication of participant pictures has been received.

**Figure 4 brainsci-13-00876-f004:**
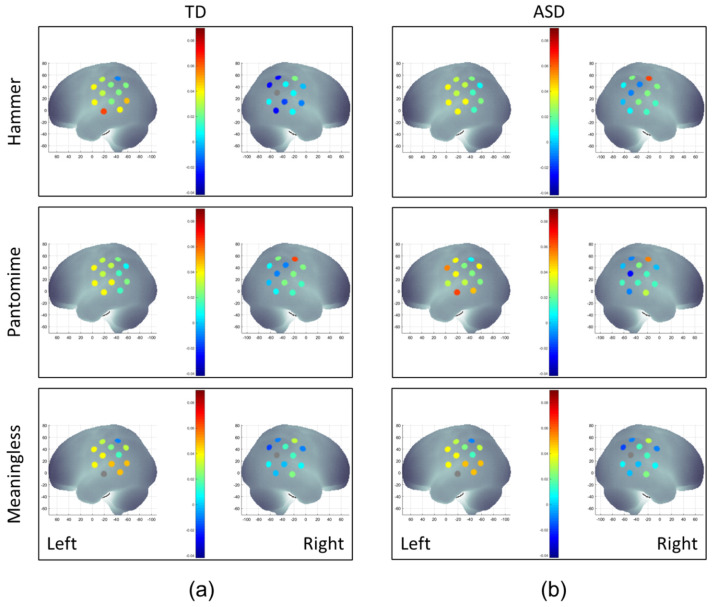
A visual representation of averaged HbO_2_ concentrations during the hammer, pantomime, and meaningless conditions in TD children without ASD (**a**) and children with ASD (**b**). HbO_2_ values on the y-axis range from −0.04, indicated by dark blue, to 0.08, indicated by red, and shades in between.

**Figure 5 brainsci-13-00876-f005:**
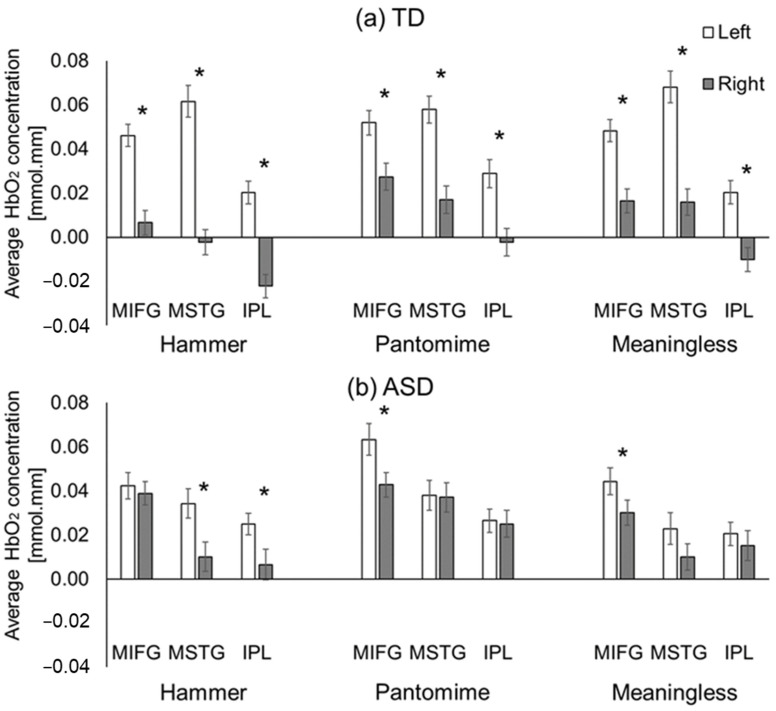
Hemispheric-related differences between TD children without ASD (**a**) and children with ASD (**b**) in average HbO_2_ concentrations. * indicates significant differences between hemispheres (*p* < 0.05).

**Figure 6 brainsci-13-00876-f006:**
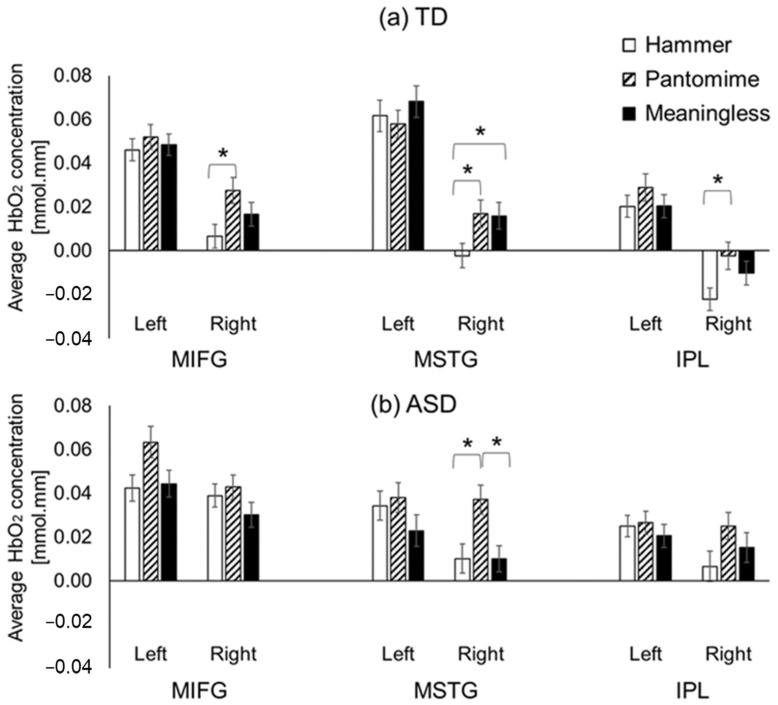
Condition-related differences between TD children without ASD (**a**) and children with ASD (**b**) in average HbO_2_ concentrations. * indicates significant differences between conditions (*p* < 0.05).

**Figure 7 brainsci-13-00876-f007:**
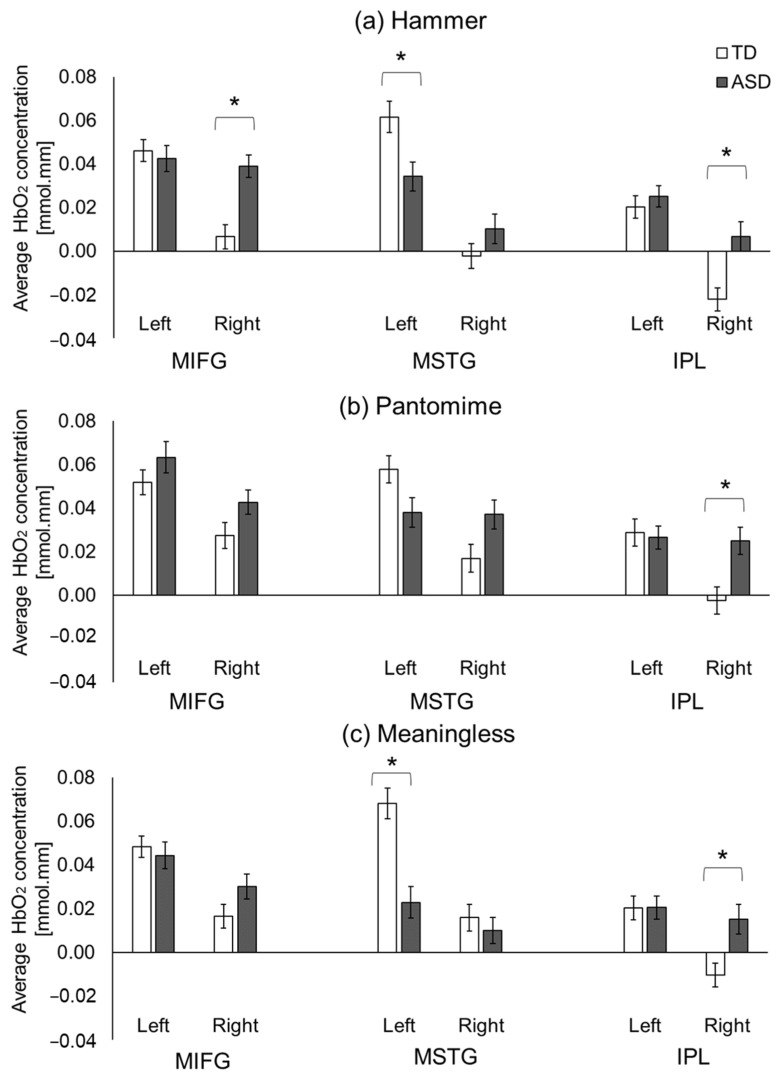
Group-related differences in average HbO_2_ concentrations during hammer (**a**), pantomime (**b**), and meaningless (**c**) conditions. * indicates significant differences between groups (*p* < 0.05).

**Table 1 brainsci-13-00876-t001:** Data on demographics and standardized measures for children with/without ASD.

Characteristics	ASD Group (N = 14) Mean ± SE	TD Group (N = 17) Mean ± SE
Age	11.14 ± 0.95	10.82 ± 0.69
Sex	9M, 5F	11M, 6F
Ethnicity	10C, 2 A, 1 AC, 1BC	13 C; 1A; 1AI; 2AC
SES-Child	64.57 ± 5.35	69.71 ± 4.43
Coren’s handedness score	Right-handed 33.36 ± 1.59	Right-handed 33.41 ± 1.78
SCQ score	23.50 ± 3.27	-
ADOS total Score Social affect Repetitive behavior	18.17 ± 1.84 13.17 ± 1.42 5.00 ± 0.62	-
VABS-II (SS)	72.50 ± 3.07 *	110.41 ± 2.91
IQ	85.64 ± 7.04 *	114.18 ± 1.71
BOT-2 manual dexterity	19.93 ± 2.04 *	27.71 ± 1.08

SES-Child = Hollingshead Four-Factor Index of Socioeconomic Status; SCQ = Social Communication Questionnaire; ADOS = Autism Diagnostic Observation Schedule—2nd Edition; IQ = Intelligence Quotient; VABS-II = Vineland Adaptive Behavior Scale—2nd Edition; BOT-2 = Bruininks–Oseretsky Test of Motor Proficiency; M = Male, F = Female, C = Caucasian, A = Asian, BC = Black–Caucasian, AC = Asian–Caucasian, AI = American Indian. * indicates significant differences between children with and without ASD (*p* < 0.05).

**Table 2 brainsci-13-00876-t002:** The correlations between cortical activation and SIPT-PP and SIPT-BMC performance.

R-Values	SIPT-PP Total Error			SIPT-BMC Total Error		
	H	P	M	H	P	M
**TD**						
Left hemisphere						
MIFG	** −0.283 ** **	** −0.296 ** **	−0.180	** −0.452 ** **	** −0.426 ** **	** −0.329 ** **
MSTG	0.002	−0.157	** 0.311 ** **	−0.091	** −0.296 ** **	0.150
IPL	0.063	−0.093	−0.042	** −0.337 ** **	** −0.336 ** **	** −0.390 ** **
Right hemisphere						
MIFG	0.025	0.064	0.019	−0.122	−0.225 *	−0.210 *
MSTG	0.111	0.098	0.192 *	−0.112	−0.103	0.030
IPL	0.129	0.087	** 0.304 ** **	−0.132	−0.169	−0.058
**ASD**						
Left hemisphere						
MIFG	** −0.257 * **	0.222 *	** 0.287 ** **	−0.072	−0.104	−0.010
MSTG	0.212 *	0.179	0.216 *	−0.032	** 0.317 ** **	** 0.332 ** **
IPL	0.160	0.095	0.190	0.105	0.213 *	** 0.330 ** **
Right hemisphere						
MIFG	** 0.434 ** **	** 0.288 ** **	** 0.283 ** **	0.186	0.153	0.049
MSTG	0.094	0.131	0.047	−0.130	0.002	−0.061
IPL	** 0.282 ** **	0.235 *	** 0.368 ** **	** 0.432 ** **	0.180	** 0.543 ** **

R-values are presented in this figure. * indicates *p* < 0.05; ** indicates *p* < 0.01. Bolded font with shading indicates that *p*-values survived for FDR corrections. H = hammer condition; P = pantomime condition; M = meaningless condition.

**Table 3 brainsci-13-00876-t003:** The correlations between cortical activation and ADOS scores.

R-Values	ADOS-SA			ADOS-RRB			ADOS-Total		
	H	P	M	H	P	M	H	P	M
ASD									
Left hemisphere									
MIFG	0.055	0.001	0.096	−0.027	−0.170	0.085	0.033	−0.056	0.103
MSTG	−0.132	** 0.310 ** **	0.209	−0.004	** 0.360 ** **	** 0.341 ** **	−0.103	** 0.361 ** **	0.277 *
IPL	0.290 *	−0.162	0.062	** 0.352 ** **	−0.068	0.237 *	** 0.343 ** **	−0.149	0.128
Right hemisphere									
MIFG	0.063	0.172	0.252 *	** 0.311 ** **	0.204	0.288 *	0.153	0.202	0.293 *
MSTG	−0.097	0.111	0.205	−0.139	0.067	−0.013	−0.122	0.109	0.155
IPL	0.045	0.214	0.072	0.217	0.142	** 0.330 ** **	0.108	0.214	0.167

R-values are presented in this figure. * indicates *p* < 0.05; ** indicates *p* < 0.01. Bolded font with shading indicates that *p*-values survived for FDR corrections. H = hammer condition; P = pantomime condition; M = meaningless condition.

## Data Availability

The data presented in this study are available on request from the corresponding author. The data are not publicly available due to restrictions associated with participants’ privacy.

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
