# Peer review of "fNIRS-Based Differences in Cortical Activation during Tool Use, Pantomimed Actions, and Meaningless Actions between Children with and without Autism Spectrum Disorder (ASD)"

_brainsci, 2023, doi:10.3390/brainsci13060876_

Round 1

Reviewer 1 Report

I found this manuscript well-written and informative. However, in my opinion, the narrative structure of the manuscript presents an unbalanced view of cognitive models. Thus, I summarized below a series of points the authors may consider revising their manuscript to enrich the theoretical argumentation. This may strengthen the current study and provide a more comprehensive understanding of the topic.

* * *

Throughout the manuscript, it appears that the authors have taken a polarizing stance in favour of grounded models of tooling (e.g., Ellis & Tucker, 2000; British Journal of Psychology). While such approaches do have some merit, it also carries the risk of oversimplifying and "under-intellectualising" cognition (e.g., Osiurak & Federico, 2021; Synthese). Scholars have suggested that the question of the embodiment is unresolvable (e.g., Michel, 2021; Phenomenology and the Cognitive Sciences). Therefore, it would be more fruitful to focus on understanding the nature of sensorimotor knowledge and how it relates to conceptual/tooling processing (e.g., Chatterjee; 2010; Language and cognition; Dove; 2011; Frontiers in Psychology; Osiurak & Federico; 2021; Synthese). Also, over two decades of neuroimaging research have shown that it is practically impossible to make a clear distinction between "motor", "sensorimotor", or "cognitive" mental activity (Meteyard et al., 2012; Cortex). Therefore, acknowledging the limitations of grounded approaches, considering alternative models, and understanding the nature of sensorimotor processing may provide a more nuanced approach to the topic. Concerning tooling, which is the main framework of the manuscript, the most recent review of Osiurak and Federico (2021; Synthese) may be helpful for the authors to consider in this regard.

Concerning the neurocognitive framework of this study, there has been a proposal by certain scholars to differentiate between visuospatial reasoning and technical reasoning in describing physical understanding, where technical reasoning involves a form of causal reasoning related to understanding properties of the physical world (Osiurak et al. 2022; Science Advances; Fischer et al., 2016; PNAS; Osiurak, Claidière & Federico; 2022 Trends in Cognitive Sciences). Critically, most recent research indicates that the inferior parietal cortex (IPC), particularly the anterior part of the supramarginal gyrus, namely, the area PF (Glasser et al., 2016; Nature), may reflect this dissociation neuroanatomically. Specifically, the left area PF of the IPC appears to be more involved in physical understanding tasks (technical reasoning), while the right area PF of the IPC appears to be more involved in visuospatial abilities (Federico et al., 2022; Scientific Reports). Therefore, the dissociation observed by the authors in their study between ASD and non-ASD participants in the right and left IPC might be informative of reduced visuospatial competence (see Caron et al., 2006; Brain; Muth et al., 2014; Journal of Autism and Developmental Disorders for putting in the context your results) in the presence of retained physical (technical) comprehension.

Globally, while the authors’ findings suggest differences in visuospatial and mentalizing abilities between ASD and non-ASD (i.e., meaningless gestures producing differences in social brain regions such as the STG), technical-mechanical functioning may remain intact. This hypothesis may be clinically intriguing and may represent a clear indication of the dissociation between visuospatial and technical-mechanical skills from basic cognitive neuroscience research. This significant result, which, based on my best knowledge, may constitute a first in the field, deserves, in my opinion, appropriate recognition and emphasis. Thus, the manuscript might benefit from further exploring and discussing these points (Federico et al., 2022; Scientific Reports; Osiurak et al., 2022; Science Advances).

Author Response

Reviewer 1:

I found this manuscript well-written and informative. However, in my opinion, the narrative structure of the manuscript presents an unbalanced view of cognitive models. Thus, I summarized below a series of points the authors may consider revising their manuscript to enrich the theoretical argumentation. This may strengthen the current study and provide a more comprehensive understanding of the topic.

Response:

  1. Throughout the manuscript, it appears that the authors have taken a polarizing stance in favour of grounded models of tooling (g., Ellis & Tucker, 2000; British Journal of Psychology). While such approaches do have some merit, it also carries the risk of oversimplifying and "under-intellectualising" cognition (e.g., Osiurak & Federico, 2021; Synthese). Scholars have suggested that the question of the embodiment is unresolvable (e.g., Michel, 2021; Phenomenology and the Cognitive Sciences). Therefore, it would be more fruitful to focus on understanding the nature of sensorimotor knowledge and how it relates to conceptual/tooling processing (e.g., Chatterjee; 2010; Language and cognition; Dove; 2011; Frontiers in Psychology; Osiurak & Federico; 2021; Synthese). Also, over two decades of neuroimaging research have shown that it is practically impossible to make a clear distinction between "motor", "sensorimotor", or "cognitive" mental activity (Meteyard et al., 2012; Cortex). Therefore, acknowledging the limitations of grounded approaches, considering alternative models, and understanding the nature of sensorimotor processing may provide a more nuanced approach to the topic. Concerning tooling, which is the main framework of the manuscript, the most recent review of Osiurak and Federico (2021; Synthese) may be helpful for the authors to consider in this regard.

Response: Thank you for your suggestions regarding the importance of both embodied approaches (e.g., Manipulation-Knowledge Hypothesis), cognitive models (Osiurak & Federico; 2021; Synthese) and sensorimotor reasoning in tool use. We have taken your feedback and have made changes to our behavioral framework, which are outlined in the introduction section under 1.1. Behavioral Framework for Tool Use and summarized in Figure 1 (Pages 2-3, Lines 56-74). Specifically, we have emphasized that successful tool use involves perceiving the potential actions of the tool and the mechanical actions between the tool and the object, such as hammering-ability or grasp-ability. According to a more embodied manipulation-knowledge hypothesis, one must refer to the action semantic system that stores functional knowledge about the tool (i.e., what the tool is used for) as well as the action knowledge system that stores movement information required to use the tool (i.e., ways or actions to grasp the tool). However, this approach did not explain how people manipulate novel tools (without previous knowledge) and the mechanical actions between the objects and the tools. To better account for the manipulation of novel tools, we have incorporated the more recent technical reasoning hypothesis, which suggests that tool-use actions are based on mechanical knowledge and requires specific forms of causal and analogical reasoning to anticipate outcomes and apply this reasoning to novel situations (Osiurak & Federico, 2021; Osiurak et al., 2022; Osiurak, Claidière, Federico, 2022). We are confident that the integration of the technical reasoning hypothesis has made our behavioral framework more comprehensive and robust.

References included are listed below:

Osiurak, F., & Federico, G. Four ways of (mis-)conceiving embodiment in tool use. Synthese. 2020, 199 (1-2),3853-3879.

Osiurak, F., Claidière, N., Bluet, A., Brogniart, J., Lasserre, S., Bonhoure, T., Di Rollo, L., Gorry, N., Polette, Y., Saude, A., Federico, G., Uomini, N., Reynaud, E. Technical reasoning bolsters cumulative technological culture through convergent transformations. Sci Adv. 2022, 8(9), eabl7446.

Osiurak, F., Claidière, N., Federico, G. Bringing cumulative technological culture beyond copying versus reasoning. Trends Cogn Sci. 2022, 27(1), 30–42.

Figure 1. Behavioral Framework and potential neural mechanisms for Tool-use

  1. Concerning the neurocognitive framework of this study, there has been a proposal by certain scholars to differentiate between visuospatial reasoning and technical reasoning in describing physical understanding, where technical reasoning involves a form of causal reasoning related to understanding properties of the physical world (Osiurak et al. 2022; Science Advances; Fischer et al., 2016; PNAS; Osiurak, Claidière & Federico; 2022 Trends in Cognitive Sciences). Critically, most recent research indicates that the inferior parietal cortex (IPC), particularly the anterior part of the supramarginal gyrus, namely, the area PF (Glasser et al., 2016; Nature), may reflect this dissociation neuroanatomically. Specifically, the left area PF of the IPC appears to be more involved in physical understanding tasks (technical reasoning), while the right area PF of the IPC appears to be more involved in visuospatial abilities (Federico et al., 2022; Scientific Reports). Therefore, the dissociation observed by the authors in their study comparing ASD and non-ASD participants in the right and left IPC might be indicative of reduced visuospatial competence (see Caron et al., 2006; Brain; Muth et al., 2014; JADD) while physical (technical) comprehension is retained.

Response: Based on the new behavioral framework adopted, we now have a broader neurocognitive framework to encompass both manipulation-knowledge and technical reasoning hypotheses and to explain the potential neural mechanisms underlying tool-use actions. Specifically, studies supporting the Manipulation-knowledge hypothesis have verified the tool use network based on the recognition of the tool (Action Semantic systems) and the planning/execution of tool-related movements (Action Knowledge system). On the other hand, studies supporting the technical reasoning hypothesis have verified the network by distinguishing between visuospatial and technical reasoning (Osiurak et al. 2022; Osiurak, Claidière & Federico; 2022; Fischer et al., 2016; Figure 1). Based on this hypothesis, we also included recent fMRI studies to explain the importance of the left frontal and parietal regions for tool use. For example, using fMRI, Fischer et al. (2016) found greater activation over frontal and parietal regions when viewing physical vs. social interactions and when making physical vs. non-physical judgments. Moreover, Federico et al. (2022) found that the cortical thickness of left but not right IPL (especially the anterior part of the supramarginal gyrus; PF region) correlated with technical reasoning performance. This suggests that left PF is involved in technical reasoning of tool use, while right PF is more involved in visuospatial processing (Federico et al. 2022). Corresponding descriptions have been added to the introduction section under 1.2. Neural Mechanisms Underlying Tool-Use (Pages 3-4, Lines 97-130). We have also taken into account the potential impact of atypical visuospatial processing in technical processing for children with ASD, which may compromise their tool-use performance (Muth et al., 2014; Caron et al., 2006). Corresponding descriptions have been added to the introduction sections under 1.3. Dyspraxia in Children with ASD (Page 4, Lines 158-160).

Reference:

Fischer, J., Mikhael, J. G., Tenenbaum, J. B., Kanwisher, N. Functional neuroanatomy of intuitive physical inference. Proc Natl Acad Sci U S A. 2016, 113(34), E5072–E5081.

Osiurak, F., Claidière, N., Bluet, A., Brogniart, J., Lasserre, S., Bonhoure, T., Di Rollo, L., Gorry, N., Polette, Y., Saude, A., Federico, G., Uomini, N., Reynaud, E. Technical reasoning bolsters cumulative technological culture through convergent transformations. Sci Adv. 2022, 8(9), eabl7446.

Osiurak, F., Claidière, N., Federico, G. Bringing cumulative technological culture beyond copying versus reasoning. Trends Cogn Sci. 2022, 27(1), 30–42.

Federico, G., Reynaud, E., Navarro, J., Lesourd, M., Gaujoux, V., Lamberton, F., Ibarrola, D., Cavaliere, C., Alfano, V., Aiello, M., Salvatore, M., Seguin, P., Schnebelen, D., Brandimonte, M. A., Rossetti, Y., Osiurak, F. The cortical thickness of the area PF of the left inferior parietal cortex mediates technical-reasoning skills. Sci Rep. 2022, 12(1), 11840.

Muth, A., Hönekopp, J., Falter, C. M. Visuo-spatial performance in autism: a meta-analysis. J Autism Dev Disord. 2014, 44(12), 3245–3263.

Caron, M. J., Mottron, L., Berthiaume, C., Dawson, M. Cognitive mechanisms, specificity and neural underpinnings of visuospatial peaks in autism. Brain. 2006, 129(Pt 7), 1789–1802.

  1. Globally, while the authors’ findings suggest differences in visuospatial and mentalizing abilities between ASD and non-ASD (i.e., meaningless gestures producing differences in social brain regions such as the STG), technical-mechanical functioning may remain intact. This hypothesis may be clinically intriguing and may represent a clear indication of the dissociation between visuospatial and technical-mechanical skills from basic cognitive neuroscience research. This significant result, which, based on my best knowledge, may constitute a first in the field, deserves, in my opinion, appropriate recognition and emphasis. Thus, the manuscript might benefit from further exploring and discussing these points (Federico et al., 2022; Scientific Reports; Osiurak et al., 2022; Science Advances).

Response: Thanks for the valuable interpretation of our findings. We have largely modified 4.4.1. Intact Left IPL but Hyperactivation over Right IPL in Children with ASD under the discussion section to better interpret our findings (Page 20, Lines 642-661). Specifically, we found that children with ASD showed typical activation over the left IPL but hyperactivation in the right IPL region during all three types of gesturing. The left IPL, particularly the anterior portion of the supramarginal gyrus, is said to be important for technical reasoning (Federico et al., 2022) while the right IPL is known to play an important role in visuospatial processing (Federico et al., 2022). Greater right IPL activation in children with ASD might reflect their atypical performance in processing visuospatial and information during tool use (Muth et al., 2014; Caron et al., 2006). Children with ASD might have intact technical reasoning (indicated by typical left IPL activation) but atypical visuospatial and proprioceptive processing (indicated by hyperactivation over right IPL), leading to insufficient tool-use performance. We also emphasized these findings in the abstract (Page 1, Lines 24-26) and in conclusion and associated clinical implications (Page 21, Lines 704-714).

Reference:

Federico, G., Reynaud, E., Navarro, J., Lesourd, M., Gaujoux, V., Lamberton, F., Ibarrola, D., Cavaliere, C., Alfano, V., Aiello, M., Salvatore, M., Seguin, P., Schnebelen, D., Brandimonte, M. A., Rossetti, Y., Osiurak, F. The cortical thickness of the area PF of the left inferior parietal cortex mediates technical-reasoning skills. Sci Rep. 2022, 12(1), 11840.

Muth, A., Hönekopp, J., Falter, C. M. Visuo-spatial performance in autism: a meta-analysis. J Autism Dev Disord. 2014, 44(12), 3245–3263.

Caron, M. J., Mottron, L., Berthiaume, C., Dawson, M. Cognitive mechanisms, specificity and neural underpinnings of visuospatial peaks in autism. Brain. 2006, 129(Pt 7), 1789–1802.

Reviewer 2 Report

1.      Quantitative results must be included in the abstract section.

2.      At the end of your abstract, please provide a "take-home" message.

3.      Keywords should have been reorganized alphabetically.

4.      I am encouraging the authors to not use abbreviations in the keywords.What is the novel bought by the authors in the current submission? Its works have been widely discussed in the past. Nothing something really new in the present form. The lack of a novel seems to make the present submission like to replication/modified work. The authors need to detail their novelty in the introduction section. It is a major concern for rejecting this paper.

5.      It is essential to summarize previous works' merits, novelties, and limitations in the introductory part to emphasize the gaps in the article that the latest effort seeks to address.

6.      Previouse related research regarding autism spectrum disorder needs to incorporated to highlight recent research progress, the relevant study as follows: Physiological Effect of Deep Pressure in Reducing Anxiety of Children with ASD during Traveling: A Public Transportation Setting. Bioengineering. 2022;9: 157. doi:10.3390/bioengineering9040157

7.      The last paragraph of the introduction section should clearly explain the objective of the present study.

8.      In the introduction section, it would improve the quality of the present work by providing an additional related figure.

-

Author Response

Reviewer 2:

  1. Quantitative results must be included in the abstract section.

Response: Thanks for the suggestion. We have included quantitative results in the abstract, including average praxis errors (Praxis error: TD = 17.9±1.7; ASD = 27.0±2.6, p<0.01) and their correlations with cortical activation (r=0.257 to 0.543) (Page 1, Lines 19-21).

  1. At the end of your abstract, please provide a "take-home" message.

Response: We have provided the take-home message at the end of the abstract (Page 1, Lines 24-29). Specifically, our findings indicate intact technical reasoning (typical left IPL activation) but atypical visuospatial and proprioceptive processing (hyperactivation over right IPL) during tool use in children with ASD. These results have important implications for clinicians and researchers, who should focus on facilitating visuospatial and proprioceptive processing in children with ASD. Additionally, fNIRS-related biomarkers could be used for early identification in relation to early tool use and to examine neural effects following gesture-based interventions.

  1. Keywords should have been reorganized alphabetically.

Response: We have reorganized the keywords alphabetically.

  1. I am encouraging the authors to not use abbreviations in the keywords.

Response: We ensured that no abbreviations were used in keywords.

  1. What is the novel bought by the authors in the current submission? Its works have been widely discussed in the past. Nothing really new in the present form. The lack of a novel seems to make the present submission like to replication/modified work. The authors need to detail their novelty in the introduction section. It is a major concern for rejecting this paper.

Response: The current study provides new insights into the neural mechanisms underlying tool-use difficulties in children with ASD in two ways: 1) we investigated neural activity during actual tool use in a naturalistic environment, which has not been examined in previous fMRI and EEG studies. 2) we utilized both traditional, grounded manipulation-knowledge and recent technical-reasoning hypotheses to interpret our findings (Figure 1). Please find more details and rationale in our answer to your question 6. Related descriptions were added to better describe the knowledge gap and the novelty of the study (Page 5, Lines 202-210).

  1. It is essential to summarize previous works' merits, novelties, and limitations in the introductory part to emphasize the gaps in the article that the latest effort seeks to address.

Response: Thanks for the suggestion. We have summarized previous fMRI and EEG findings, their limitations/knowledge gaps, and the novelty of the current study in 1.4. ASD-Related Atypical Cortical Activation During Tool Use and Pantomimed of Tool Use and 1.5. Aims and Hypotheses of This Study sections (Pages 5-6, Lines 168-216). Specifically, although previous fMRI and EEG studies suggested potential neural mechanisms underlying the gesturing difficulties in children with ASD (summarized in Page 5, Lines 182-191), they didn’t include actual tool use and were usually conducted in structured, restricted environments. This might be due to the constraint of neuroimaging tools. For instance, fMRI requires participants to lie still in the scanner, which may not be suitable for children, while EEG has poor spatial resolution and cannot provide structural information. In contrast, fNIRS is a non-invasive neuroimaging technique with greater temporal resolution than fMRI and greater spatial resolution than EEG. More importantly, its post-processing accommodates movement artifacts and constrains the subject only by a cap, making it an ideal neuroimaging tool for children with ASD. Given the advantage of fNIRS, the current study expanded the paradigm to investigate neural activity during actual tool use. Furthermore, previous studies interpreted their results based on a grounded, embodied approach  (e.g., manipulation-knowledge hypothesis) without considering the reasoning process during tool use. In the current study, we use both traditional, grounded manipulation-knowledge and recent technical-reasoning hypotheses to interpret our findings.

  1. Previous related research regarding autism spectrum disorder needs to be incorporated to highlight recent research progress, the relevant study as follows: Physiological Effect of Deep Pressure in Reducing Anxiety of Children with ASD during Traveling: A Public Transportation Setting. Bioengineering. 2022;9: 157. doi:10.3390/bioengineering9040157

Response: We highlight that anxiety and other behavioral symptoms can have an impact on children's tool-use performance and training outcomes. Therefore, it is important for clinicians and future researchers to use techniques such as deep pressure to manage ASD-related symptoms (Afif et al., 2022). Related descriptions have been added on Page 21, Lines 715-718.

Reference:

Afif, I. Y., Manik, A. R., Munthe, K., Maula, M. I., Ammarullah, M. I., Jamari, J., Winarni, T. I. Physiological effect of deep pressure in reducing anxiety of children with ASD during traveling: a public transportation setting. Bioengineering. 2022, 9(4), 157.

  1. The last paragraph of the introduction section should clearly explain the objective of the present study.

Response: We have specified our objective in the last paragraph of the introduction section (Page 5, Lines 204-210). Specifically, the objective of the current study is to investigate cortical activation in children with and without ASD during actual tool use, pantomimed tool-use, and when performing meaningless actions with similar movement kinematics. Moreover, we aim to correlate tool-related cortical activation with standardized praxis measures and the ASD severity of children with ASD.

  1. In the introduction section, it would improve the quality of the present work by providing an additional related figure.

Response: Thanks for the suggestion. We have included a figure in the introduction section to better summarize our behavioral and neurocognitive frameworks underlying tool-related actions. Pleases see details in Figure 1.

Round 2

Reviewer 1 Report

The authors have effectively addressed all the concerns I raised. In light of the ongoing discussion on technical reasoning versus manipulation knowledge, the manuscript now presents a more comprehensive theoretical framework, which I believe adequately also addresses the principal concern the second anonymous reviewer expressed. Additionally, the manuscript highlights a significant aspect regarding intact technical reasoning and impaired visuospatial abilities in individuals with ASD. This, based on my best knowledge, is a very first in the literature. Considering these factors, I strongly recommend publishing this work and extend my heartfelt congratulations to the authors for their contribution.

In Figure 1: 

Left IPF, especially the PF regions are , is important for technical reasoning